# Etiology of Foliar Blight of Indian Paintbrush (*Castilleja tenuiflora*) in Mexico

**DOI:** 10.3390/microorganisms12081714

**Published:** 2024-08-20

**Authors:** Alma Rosa Solano-Báez, Gabriela Trejo-Tapia, Miroslav Kolařík, Jossue Ortiz-Álvarez, José Luis Trejo-Espino, Guillermo Márquez-Licona

**Affiliations:** 1Instituto Politécnico Nacional, Centro de Desarrollo de Productos Bióticos, Yautepec 62731, Morelos, Mexico; asolanob@ipn.mx (A.R.S.-B.); gttapia@ipn.mx (G.T.-T.); jtrejo@ipn.mx (J.L.T.-E.); 2Institute of Microbiology, Czech Academy of Sciences, CZ-142 20 Prague, Czech Republic; miroslavkolarik@seznam.cz; 3Research Program: “Investigadoras e Investigadores por México” Consejo Nacional de Humanidades, Ciencias, Tecnologías e Innovación (CONAHCyT), Ciudad de México 03940, Mexico; jossue.ortiz@conacyt.mx

**Keywords:** *Alternaria*, morphology, pathogenicity, phylogeny, Mexican medicinal plants, Orobanchaceae

## Abstract

*Castilleja tenuiflora* is a native perennial plant used in traditional Mexican medicine. In June 2022, leaf blight symptoms were observed in a wild population of *C. tenuiflora* plants. Disease incidence was 80% and disease intensity reached up to 5% of the leaf area. Currently, there are no reports of pathogens causing leaf blight in this plant; therefore, this work aimed to identify the fungi responsible for the disease. The fungi recovered from the diseased tissue were characterized by means of pathogenicity tests and cultural, morphological, and molecular characterization. The information obtained revealed that *Alternaria alternata* and *Alternaria gossypina* are the pathogens responsible for the disease. This is the first report implicating species of *Alternaria* in causing leaf blight of *C. tenuiflora* in Mexico, as well as the first report of *Alternaria gossypina* also in Mexico. These pathogens may threaten the in situ conservation of native *C. tenuiflora* populations and limit their in vitro propagation. Future research lines should focus on determining the effect of these pathogens on metabolite production.

## 1. Introduction

*Castilleja tenuiflora* (Orobanchaceae), commonly known in Mexico as “Hierba del cáncer” [1,2,3], “Garañona” [1], “Yerba de la epístola” [4], “Cola de borrego” [5], and “Atzoyatl” [in Nahuatl] [5], is a small perennial plant native to the southern of the United States and Mexico that thrives in pine–oak forests [6]. This medicinal plant has been traditionally used in Mexican medicine for its pharmacological properties, including cancer treatment, as a remedy for gastric ulcers, respiratory disorders [7,8,9,10], and depression [11], and as an anti-inflammatory agent [6].

Given the pharmacological importance of *C. tenuiflora*, it is essential to preserve its native populations. An important part of this is the identification of diseases that may threaten its conservation in situ [12,13], its propagation in vitro, or even the production of the metabolites of interest [14,15,16]. The study of the diseases affecting this plant is limited, and so far only two rusts (fungi from the order Pucciniales) have been reported, (a) *Puccinia nessodes* in Mexico [17] and Guatemala [18] and (b) *Cronartium colesporoides* in Guatemala [18] and Costa Rica [19]. Therefore, there is a lack of information on foliar diseases that may occur in wild *C. tenuiflora* plants.

*Alternaria* is a ubiquitous fungal genus of the family Pleosporaceae [20,21,22] with more than 4000 species [23], including endophytic, saprophytic, opportunistic human-pathogenic, and plant-pathogenic species associated with a wide variety of hosts [24]. The plant-pathogenic *Alternaria* species are capable of infecting leaves, stems, roots, flowers, and fruits of vegetables, ornamentals, and trees. Diseases caused by *Alternaria* species are characterized by the induction of symptoms of leaf spots and blights, but they may also cause damping-off and stem, tuber, and fruit rot, as one of the genera of plant-pathogen fungi that generate the most losses [25]. The distinctive morphological feature of the genus *Alternaria* is the production of dark brown chains of phaeodictyospores [24]. However, morphological similarities between *Alternaria* species do not allow a properly morphological-based identification. Therefore, a multifaceted identification encompassing cultural, morphological, molecular, and phylogenetic identification is recommended [24].

During a plant collection trip to study the metabolites produced by wild *C. tenuiflora*, leaf blight symptoms were observed. Therefore, the objective of this study was to identify the fungi responsible for these symptoms in *C. tenuiflora* plants from the Iztaccíhuatl-Popocatépetl National Park in Mexico. 

## 2. Materials and Methods

### 2.1. Sample Collection, Isolation, Purification, and Preservation of the Fungus Associated with Indian Paintbrush Leaf Blight 

Twelve diseased plants of *C. tenuiflora* were collected at the Iztaccíhuatl-Popocatépetl National Park (latitude N 19°05′20″, longitude W 98°40′25.3″, at 3446 m.a.s.l) in June 2022. The sampled branches were placed in labeled paper bags and then enclosed in plastic bags. The collected samples were stored and transported in a cooler for processing at the Phytopathology Laboratory of the Center of Development of Biotic Products of the *Instituto Politécnico Nacional* (CEPROBI-IPN, by its acronym in Spanish). 

For the isolation of fungi associated with the disease, the samples were sub-sampled to obtain leaf fragments of approximately 0.5 cm^2^ from the transition zone between healthy and diseased tissue. Tissue sections were disinfested with 1.5% sodium hypochlorite for 90 s, followed by a triple wash with sterile distilled water. The disinfested plant tissue was dried for 24 h between sterile blotting paper. The plant tissue fragments were then transferred to Petri dishes containing an acidified Potato Dextrose Agar (PDA; Bioxon, Mexico) medium and incubated in the dark at 28 °C for 48 h. Mycelial growth from the tissue sections was transferred to Petri dishes with a 2% Water–Agar (WA) medium and incubated in the dark at 28 °C for 24 h. 

To obtain axenic isolates, a single spore was collected from each isolate using a stereoscopic microscope and an insulin needle. The spores were transferred back to Petri dishes with the acidified PDA medium and incubated in the dark at 28 °C until the fungus filled the plates. A total of 24 isolates were recovered from the diseased leaves of *C. tenuiflora* plants. The axenic isolates were grouped by morphotype, and one isolate per morphotype was randomly selected. Based on a microscopic examination of the morphological characteristics of the conidia, all isolates were tentatively delineated as *Alternaria*. For morphological and molecular characterization, three isolates were randomly selected, one for each morphotype observed. The isolates were stored in 10% glycerol and incorporated into the Culture Collection of Phytopathogenic Fungi of the CEPROBI-IPN under the accession numbers IPN 10.0151, IPN 10.0152, and IPN 10.0153.

### 2.2. Cultural and Morphological Characterization

For cultural characterization, one isolate of each morphotype was grown in V-8 juice agar (V-8) to stimulate pigment production and in potato–carrot agar (PCA) to stimulate the development of growth rings in addition to pigment production at 25 ± 2 °C for 7 days in a daily fluorescent light/dark cycle of 8/16 h [26]. To determine the growth rate, mycelial plugs 5 mm in diameter were placed at the center of plastic plates (90 × 15 mm) containing the PDA medium and incubated under the same conditions. The colony radius of three replicates for each isolate was documented every 24 h for 7 days. The growth rate was estimated and expressed in mm day^−1^ [27]. This experiment was conducted twice. 

For morphological description of reproductive structures, isolates were grown on synthetic nutrient-poor agar (SNA) plates with a small piece of autoclaved filter paper on the agar surface [24]. The plates were incubated at 24 ± 2 °C for 7 days in a daily fluorescent light/dark cycle of 8/16 h without humidity control in unsealed plates. After the incubation, semi-permanent preparations were obtained by using lactic acid as a mounting medium; the preparations were sealed with a wax ring. From each isolate, 50 conidia were observed and measured using an Axio Imager.A2 microscope (Zeiss, Oberkochen, Germany). 

### 2.3. DNA Extractions

Total genomic DNA was obtained from fresh fungal mycelia. Biomass from each isolate was obtained by growing pure colonies on PDA for four days and incubating them in the dark at 25 °C. Mycelia were recovered from the media by using a sterile spatula and placed in a sterile microtube. DNA was extracted using the DNeasy Plant Mini Kit (Qiagen, Valencia, CA, USA) according to the manufacturer’s protocol. DNA integrity was evaluated on a 1% agarose gel and the amount of DNA was determined using a Nanodrop^®^ One (Thermo Fisher Scientific, Madison, WI, USA). All samples were diluted in sterile MB grade nuclease-free water, adjusting the DNA concentration to 10 ng/μL and storing them at −20 °C.

### 2.4. Polymerase Chain Reactions (PCR) and Sequencing

The Internal Transcribed Spacer region (ITS) and the partial sequences of the glyceraldehyde-3-phosphate dehydrogenase (gapdh), RNA polymerase II (rpb2), and translation elongation factor 1α (tef1) genes were amplified using the following primer pairs: ITS5/ITS4 [28], gpd1/gpd2 [29], RPB2-5F2 [30]/fRPB2-7cR [31], and EF1-728F/EF1-986R [32], respectively. The PCR master mix of each isolate was performed for each molecular marker, using 50 μL of the total mix volume, consisting of 25 μL of PCR Master Mix 2X (Promega Corporation, Madison, WI, USA), 10 μM of each primer, 16 μL of nuclear-free water, and 4 μL of template DNA. PCR amplification for ITS, *tef1*, and *rpb2* was carried out in a Piko Thermal Cycler (Thermo Fisher Scientific, Waltham, MA, USA), using the parameters previously described by Woudenberg et al. (2013) [24], while *gapdh* was amplified according to the protocol developed by Berbee et al. (1999) [29]. To estimate the size and the integrity of the amplicons, electrophoresis on 1% agarose gel was conducted at 120 volts for 30 min employing a 100 pb DNA ladder as a reference (Promega, Madison, WI, USA). PCR products were stained with ethidium bromide and visualized in a photodocumentation system (UVP, Thomas Scientific, Swedesboro, NJ, USA). The amplified PCR products were purified by using the MinEliute^®^ Gel Extraction Kit (Qiagen, Valencia, CA, USA) following the manufacturer’s instructions. For each amplicon, the sequencing of forward and reverse sequences was performed by Macrogen (Macrogen Inc., Seoul, Republic of Korea). The consensus sequences were generated using the UGene software [33] and were deposited in the GenBank database of the National Center for Biotechnological Information (Table 1).

### 2.5. Identification by Phylogenetic Reconstruction

The partial sequences of *gapdh*, *rpb2*, and *tef*, as well as the ITS of each isolate, were subjected to compassion analysis in the non-redundant GenBank database via BLASTn (https://blast.ncbi.nlm.nih.gov/Blast.cgi) to identify closely related taxa. Taxonomically related reference sequences were collected considering the presumptive assignation of species criteria via preliminary analysis of the ITS marker (≥80% of query coverage and ≥97% of similarity percentage) [34,35]. Sequences from the study were visualized and manually edited with Seaview, version 5 [36], if they displayed contaminations or erroneous nucleotides. Once downloaded, the sequences were aligned in MAFFT version 7 [37] and trimmed in Mega V7 [38]. Then, the sequences were concatenated in Mesquite V3.7 [39]. Phylogenetic reconstruction was performed based on sequences belonging to closely related species to our studied strains, previously described and analyzed by Woudenberg et al. (2015) [40].

For phylogenetic reconstruction, a Maximum Likelihood (ML) analysis was performed with RaxmlGUI 2.0 [41] using the GTR+G model. The ML phylogenetic tree was inferred with a Bootstrap analysis of 1000 replicates for branch support. Additionally, a Bayesian Inference (BI) analysis was performed using MrBayes v.3.1.2 [42] with a specific substitution model for each partition selected based on the Akaike Information Criterion (AIC) and the Bayesian Information Criterion (BIC). Markov Chain Monte Carlo (MCMC) with 500,000 generations was run to determine posterior probabilities. Trees were sampled every 1000 generations, and 25% of the trees displayed in the burn-in phase were discarded. Phylogenetic trees were edited and visualized using FigTree (http://tree.bio.ed.ac.uk/software/figtree/).

### 2.6. Pathogenicity Tests

Pathogenicity tests were conducted independently on the three isolates. Pathogenicity was demonstrated by spraying 1 mL of a spore suspension (1 × 10^4^ conidia mL^−1^) on the leaves of four 100-day-old cultivated plants of *C. tenuiflora* [43]. Two plants of *C. tenuiflora* were sprayed with sterile distilled water and used as negative control plants. The inoculated plants were incubated at 22 ± 2 °C for 2 days in a daily fluorescent light/dark cycle of 8/16 h without humidity control in semi-sealed vitro boxes. The experiment was repeated twice. The presence or absence of leaf blight symptoms in the plants served to determine whether the isolate was pathogenic or not.

## 3. Results

The initial symptoms of the disease manifested as light brown spots that gradually spread across the leaf blade, which became necrotic. The foliar blight was first observed as a dark brown in color before turning black, with the presence of concentric rings as seen in other hosts [44,45,46,47]. The disease incidence in the *C. tenuiflora* plants in the Iztaccíhuatl-Popocatépetl National Park was 80% (*n* = 50 evaluated plants), with a disease intensity of up to 5%.

### 3.1. Cultural and Morphological Characterization

The colonial morphology of the isolate IPN 10.0151 on PDA was slightly convex with dense aerial mycelium; colonies were initially gray, becoming olive green with sporulation, whereas on medium V-8, the colony was flat grayish with whitish mycelium at the edges. Sporulation on V-8 was moderate and scattered, with a moderate amount of nonsporulating aerial hyphae in the colony. Sporulation on PCA was abundant, with minor nonsporulating aerial hyphae except in the lighter rings of growth produced during the dark phases of the photoperiodic cycle, and over time, four pairs of concentric rings of growth and sporulation were developed. The growth rate was variable in the different culture media employed, with PDA growth rates of 3.2 mm/day, PCA 4.3 mm/day, and V-8 3.9 mm/day. Primary conidiophores measured up to 59.2 µm (mean 29.7 × 2.3 µm) in length. Conidia were olive-brown phaeodictyospores, narrow-ovoid or ellipsoid, measuring 17.3 × 6.6 µm (Figure 1).

Colonies of the isolates IPN 10.0152 and IPN 10.0153 on PDA were umbonate with dense aerial mycelium; colonies were initially grayish dark to olive green, although the isolate IPN.10.0153 had a more intense pigmentation with abundant sporulation in the center. In V-8, the colonies were flat grayish white for isolate IPN 10.0152 to dark olive green with a white center and abundant sporulation in the isolate IPN 10.0153. Sporulation on PCA was abundant, with the presence of lighter rings produced during dark phases of the photoperiod cycle, which were more evident in the isolate IPN 10.0153, and over time, the growth of four pairs of concentric growth rings and sporulation were observed in both isolates but with more abundant sporulation in isolate IPN 10.0153. The growth rate was variable in the three culture media, PDA 4.9 mm/day, PCA 5.5 mm/day, and V-8 4.7 mm/day. Primary conidiophores were up to 42 µm (mean 29.7 × 2.4 µm). Conidia were phaeodictyospores; first conidia were long-elliptical, 17.8 × 5–7 µm, conidia produced later in the chain became ovoid, ellipsoid, or subsphaeroid (Figure 1).

The morphological features of isolates IPN 10.0151, IPN 10.0152, and IPN 10.0153 tentatively placed them within the genus *Alternaria,* section *Alternaria* [26,40]. Isolate IPN 10.0151 showed reduced colonial growth compared to isolates IPN 10.0152 and IPN 10.0153, especially in the V8 agar and PDA media (Figure 1).

### 3.2. Phylogenetic Identification

Since microscopic and cultural characterization are not sufficient determinants for the assignment of species belonging to *Alternaria*, taxonomic classification and identification must be complemented by the employment of phylogenetic reconstructions based on key molecular markers such as *gapdh*, *rpb2*, *tef1*, and ITS [40], which are used to perform an accurate species delimitation of *Alternaria* [48]. The presumptive identification-based analysis of ITS by BLASTn exhibited a ≥97% identity of the study strains with *Alternaria alternata* and *A. gossypina*. Optimization of the ML tree revealed a likelihood value of −4492.765252. The alignment was composed of 136 patterns. The ML tree was constructed based on the following empirical frequencies: freq pi(A) = 0.244842, freq pi(C) = 0.272964, freq pi(G) = 0.250118, and freq pi(T): 0.232076; a rate parameter of A-G = 3.683673, C-T= 8.267649; plus an alpha distribution of alpha: 0.089339. In the case of the BI tree, the Bayesian Posterior Probabilities resulted in a final average standard deviation of the slit sequences of around 0.250000.

Phylogenetic analyses of the concatenated sequences of *gapdh*, *rpb2*, *tef1*, and *ITS* showed that the isolates IPN 10.0152 and IPN 10.0153 were clustered in *Alternaria* section *Alternata*, showing a close phylogenetic relationship with strains of *Alternaria alternata,* while isolate IPN 10.0151 was also clustered in *Alternaria* section *Alternata*, but this isolate was closely related to *Alternaria gossypina* (Figure 2). 

### 3.3. Pathogenicity Tests

All three *Alternaria* isolates were pathogenic to *C. tenuiflora* plants. Regardless of the isolate inoculated, the plants developed brown to black blights on the inoculated leaves and stems (Figure 3) with abundant mycelium 4 days after inoculation (DAI), and the blights progressed to complete necrosis of the plants after 7 days. The symptoms observed in the inoculated plants were the same as those observed in the naturally infected plants, from which the fungi were originally isolated. Moreover, the control plants remained asymptomatic throughout the experiment. The fungi were re-isolated from the symptomatic plants and found to be morphologically identical to the inoculated fungi, thus fulfilling Koch’s postulates. All three isolates induced similar symptoms in the inoculated plants. The results indicate that *A. alternata* and *A. gossypina* are the causal agents of leaf blight in *C. tenuiflora* plants growing in the Iztaccíhuatl-Popocatépetl National Park. 

## 4. Discussion

Disinfestation of diseased plant tissues with sodium hypochlorite is a common method for isolation of phytopathogenic fungi and bacteria, as it eliminates microorganisms present on the surface of the sample, allowing only the recovery of disease-associated organisms present within the plant tissue of interest [49]. However, it has been observed that the efficacy of disinfectant molecules depends on the fungal species encountered and their intraspecific diversity, as well as the physiological state of the spores [50]. Similarly, fungi are more resistant to chlorine inactivation than bacteria and viruses; however, this resistance or sensitivity varies depending on the genus and species of the fungus in question [51]. In this study, the disinfestation process successfully allowed the recovery of the causal agents of the disease, without the detection of other microorganisms during the isolation process.

Due to the traditional use of *C. tenuiflora*, the study of diseases affecting this plant species is limited, in contrast to economically important crops. However, the interest in identifying diseases affecting this plant arises from the potential for intensive production, given recent discoveries regarding the production of metabolites of pharmacological interest. Thus, it is expected that in the coming years, the study of biotic and abiotic factors of this and other medicinal plants that may limit their propagation and cultivation will be completed, facilitating the commercial exploitation of these species. Prior to this investigation, only two species of rust-infecting *C. tenuiflora* had been reported, so the description of *Alternaria alternata* and *Alternaria gossypina* as causal agents of leaf blight represents a significant contribution to the understanding of biotic factors limiting the production of *C. tenuiflora*.

Unfortunately, given the location of the collection site where the disease was originally found, precise meteorological data are lacking. However, the meteorological conditions of the nearest community show an average temperature of 23 to 25 °C with no rainfall during the sampling period. The plant samples were collected just before the onset of the rainy season, so the plants were under water stress, a factor associated with increased plant susceptibility to diseases [52]. However, even though the plants used in the pathogenicity test were not under water stress, they still became diseased, demonstrating that water stress is not a determinant in the susceptibility of *C. tenuiflora* plants to leaf blight. Koch’s postulates were carried out under conditions of high relative humidity, a temperature of 22 ± 2 °C, and a daily fluorescent light/dark cycle of 8/16 h, which adequately simulated the conditions under which natural infection occurs and allowed us to successfully reproduce the disease. 

Species of the genus *Alternaria* are globally distributed and occupy a variety of ecological niches. As a result, species within this genus may exhibit saprophytic, endophytic, or pathogenic lifestyles in soil, air, plant, or animal tissues [53]. In this case, isolates of *Alternaria alternata* and *Alternaria gossypina* were able to infect the foliar tissue of inoculated plants, demonstrating their phytopathogenic capacity according to Koch’s postulates. While both pathogens cause foliar blight with no apparent difference in the symptoms, they induce necrosis and reduce the photosynthetic capacity of affected plants. The effect of this damage on leaf senescence and the production of metabolites that confer pharmacological properties to the plant remains unclear.

The amplified regions enriched the taxonomic analysis and allowed us to confirm the species assignment. In fact, all the results indicated that the three isolates belong to the genus *Alternaria* section *Alternata*. Isolates IPN 10.0152 and IPN 10.0153 were easily distinguished with molecular information as members of the *A. alternata* clade, while the isolate IPN 10.0151 was closely related to *A. gossypina* and *A. longipes*. Nevertheless, both ML and BL allowed the separation and resolution of the phylogenetic position of isolate IPN 10.0151 and its closely related species. Therefore, the present study concluded that the isolate IPN 10.0151 can be recognized as *A. gossypina*. According to some authors [40,54,55], *A. gossypina* is synonymized under three main pathotypes: *Alternaria colombiana, Alternaria grisea*, and *Alternaria tangelonis*. In this work, the placement of isolate IPN 10.0151 in the logipes clade shows a narrow clustering with the above pathotypes. Instead, we observed and highlighted the formation of a new branch. This finding, along with the fact that this species infects a host not previously reported, suggests that this isolate may be a new pathotype, but more robust molecular, morphological, and microscopic assay experiments are needed to confirm this assertion.

Further research is needed on the phylogenetic delimitation of the species that constitute the *Alternaria* section *Alternata*, especially between the species *A. gossypina* and *A. logipes*. According to recent trends in phylogenetic species determination [56], we suggest that genome sequencing and phylogenomic analysis using type and reference isolates will be essential in the short term since phylogenetic analyses based on multilocus typing have not allowed us to better resolve the identity of some clades within the *Alternata* section.

We isolated two strains identified as *A. alternata* according to their phylogenetic placement. Although these two strains exhibited identical morphological traits, the strains were clustered in different clades. These strains also display morphological variations, in both V8 and PDA agar. It is noteworthy that the two strains isolated in this study are phenotypically and genotypically distinct despite being derived from the same sample, but preliminary studies describe that A. *alternata* can form genetically diverse populations in the same host [57]. Although the aforementioned studies were conducted on samples obtained from geographically distinct locations, the findings of our study are very interesting in that *A. alternata* can form genotypically diverse populations in the same geographic location. Perhaps these variations are related to how these strains infect and evolve into *C. tenuiflora*. 

The *Alternaria* species reported in the present study are characterized by the induction of leaf spots and blight in several plant species. *A. alternata* has been reported to cause symptoms of blight in tomato [58,59], alfalfa [60], cotton [61], broccoli [62], potato [59], etc., while *A. gossypina* has been reported to induce leaf spot and boll rot in cotton [63].

In Mexico, *A. alternata* has been reported to infect “xoconostle” (*Opuntia matudae*) [64], broccoli [65], and onion [66]. Conversely, there are no previous reports describing plant diseases associated with *A. gossypina* in the Mexican territory; moreover, this study confirms the presence of *A. gossypina* in Mexico. There are few reports of *A. gossypina* infecting plants; the reports associate this fungus with native plants, tomatoes, and potatoes in China [59,67]. This is the first time that *A. alternata* and *A. gossypina* have been reported as plant pathogens in *C. tenuiflora*. This finding expands the host diversity that *Alternaria alternata* can infect.

Pathogenicity tests performed in potatoes and tomatoes have reported that *A. alternata* is capable of causing disease symptoms after 7–8 days of incubation [68,69]. In this study, the symptoms of infection caused by *Alternaria* spp. in *C. tenuiflora* were observed as early as 4 days after inoculation and necrosis 7 days after inoculation. The results reflect an accelerated process of infection and dissemination in the host. Perhaps our strains of study may harbor more effective mechanisms of virulence, but it is important to consider the immune response of *C. tenuiflora,* which may be less effective than the vegetable host.

Information on fungi affecting forest species in Mexico is limited or incomplete, and this lack of information on fungal diversity in national parks and reserves is even greater. There is an urgent need to obtain information on the diversity of these organisms in order to generate scientifically based and efficient preservation actions [70]. To date, *Alternaria* spp. including *A. alternata* have been reported in Mexico, causing foliar spots on *Quercus*, *Pseudotsuga macrolepis*, *Pinus*, *Cupresus*, *Eucalyptus,* etc. [71], indicating the presence of *Alternaria* species in the region near the *C. tenuiflora* collection site. There are no current reports of fungi infecting medicinal plants in the Iztaccíhuatl-Popocatépetl National Park. Therefore, because *C. tenuiflora* is used for medical treatments in several Mexican communities near the Iztaccíhuatl-Popocatépetl National Park, it is very important to maintain an innocuous preservation and propagation of native medicinal plants.

## 5. Conclusions

This is the first time that *A. alternata* and *A. gossypina* are reported as pathogens of *C. tenuiflora*. Remarkably, the results suggest that *C. tenuiflora* can be infected by genetically diverse strains of *A. alternata*. The onset of disease symptoms in *C. tenuiflora* caused by *Alternaria* spp. is observed in shorter periods compared to previous studies reported for vegetables. *A. alternata* and *A. gossypina* may jeopardize the in situ conservation of native *C. tenuiflora* populations under certain environmental conditions. Due to their ubiquitous nature, these fungi could infect other native forest plants of this National Park and pose a phytosanitary threat. In view of the pharmacological importance of *C. tenuiflora* and in order to preserve wild populations, in vitro propagation is recommended.

## Figures and Tables

**Figure 1 microorganisms-12-01714-f001:**
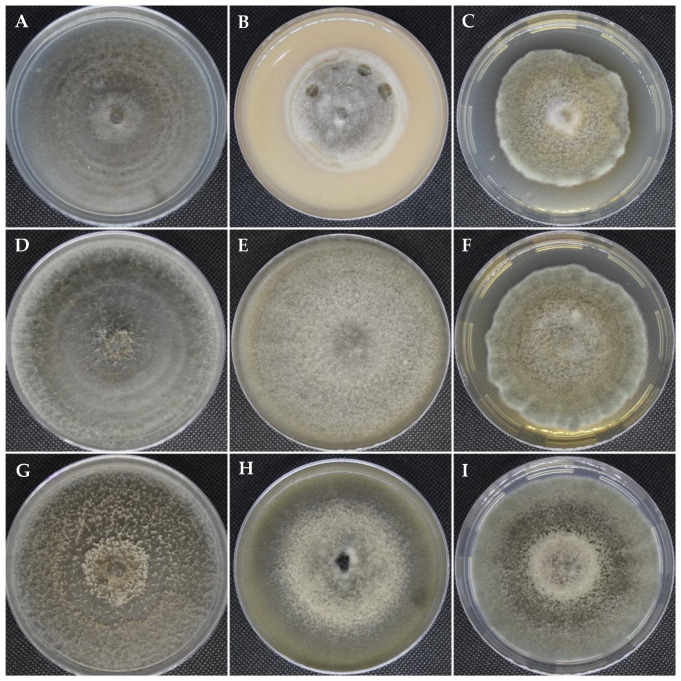
Cultural characteristics of *Alternaria* spp. isolated from *C. tenuiflora*. (**A**–**C**). Isolate IPN 10.0151 (*A. gossypina*). (**D**–**F**). Isolate IPN 10.0152 (*A. alternata*). (**G**–**I**). Isolate IPN 10.0153 (*A. alternata*). (**A**,**D**,**G**). Colonies on PCA, 10 d growth at 25 °C. (**B**,**E**,**H**). Colonies on V8 Agar, 10 d growth at 25 °C. (**C**,**F**,**I**). Colonies on PDA, 10 d growth at 25 °C.

**Figure 2 microorganisms-12-01714-f002:**
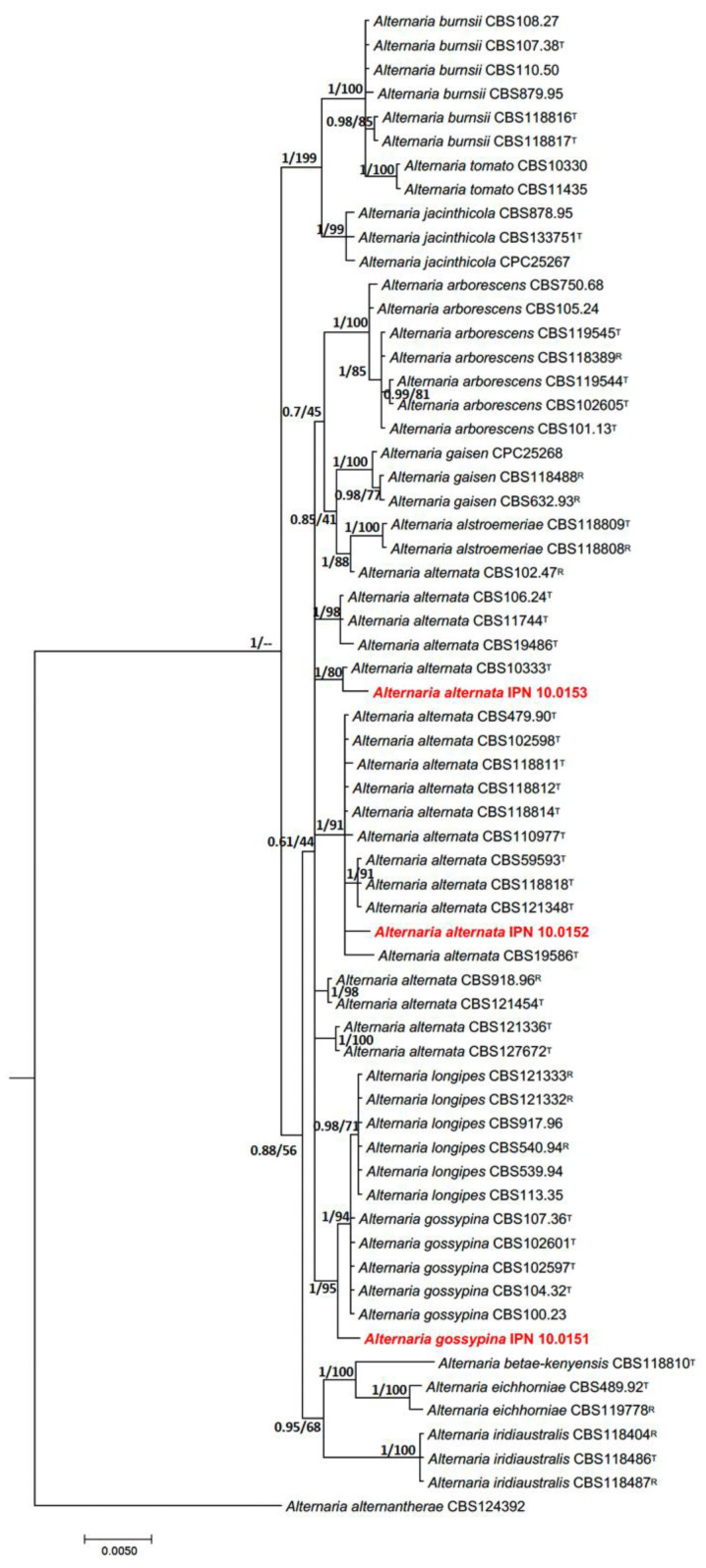
Phylogenetic tree of *Alternaria* strains recovered from *C. tenuiflora* plants. The phylogeny was constructed based on *gapdh*, *rpb2*, and *tef-1* partial gene sequences and ITS. Numbers at nodes in bold represent the Posterior Probability of Bayesian inference followed by ML Bootstrap support values. Branch lengths are proportional to the number of substitutions per site (scale bar). T: ex-type isolate; R: representative isolate. Isolates from this study are indicated in red.

**Figure 3 microorganisms-12-01714-f003:**
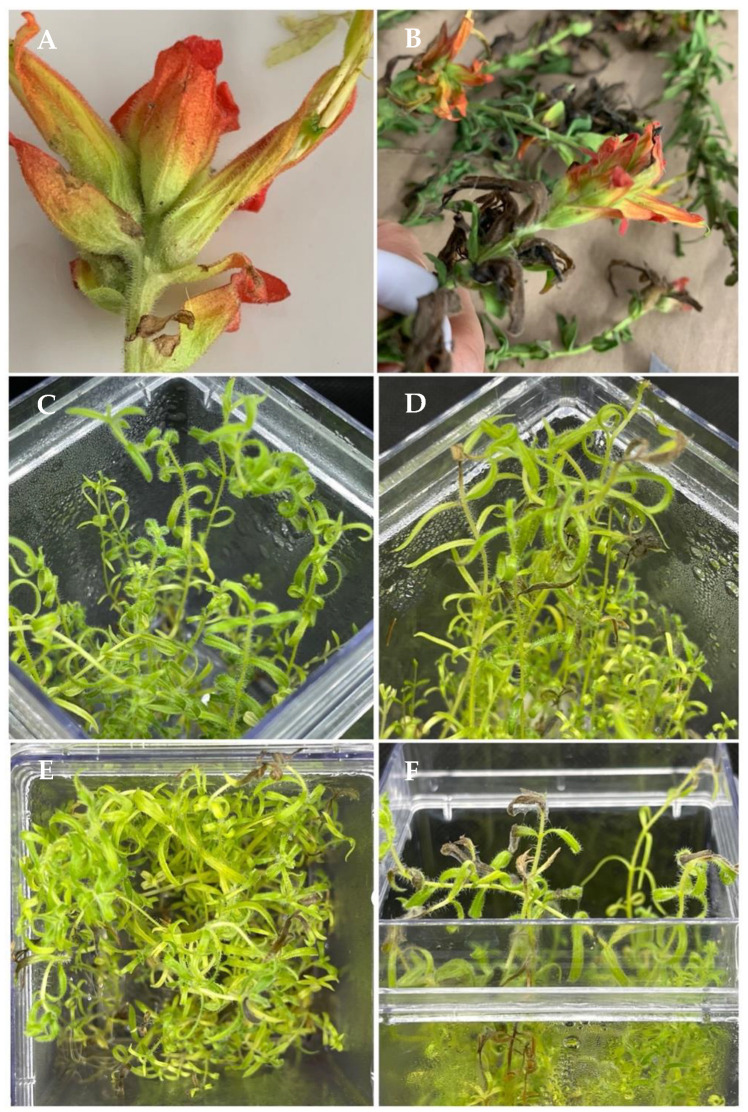
Foliar blight on *Castilleja tenuiflora* plants. (**A**) Symptoms of blight on petals. (**B**) Symptoms of leaf blight. (**C**) Healthy *C. tenuiflora* plant in vitro (control, non-inoculated plant). (**D**) Foliar blight caused by *Alternaria gossypina* (isolate IPN 10.0151). (**E**) Foliar blight caused by *Alternaria alternata* (isolate IPN 10.0152). (**F**) Foliar blight caused by *Alternaria alternata* (isolate IPN 10.0153).

**Table 1 microorganisms-12-01714-t001:** Isolates of *Alternaria* section *Alternata* used in this study for phylogenetic analyses and their GenBank accession numbers.

Species	Isolate Code	ITS	*gapdh*	*rpb2*	*tef1*
*Alternaria alstroemeriae*	CBS 118808	KP124296	KP124153	KP124764	KP125071
	CBS 118809	KP124297	KP124154	KP124765	KP125072
*Alternaria alternata*	CBS 106.24	KP124298	KP124155	KP124766	KP125073
	CBS 103.33	KP124302	KP124159	KP124770	KP125077
	CBS 117.44	KP124303	KP124160	KP124772	KP125079
	CBS 102.47	KP124304	KP124161	KP124773	KP125080
	CBS 194.86	KP124316	KP124172	KP124784	KP125092
	CBS 195.86	KP124317	KP124173	KP124785	KP125093
	CBS 479.90	KP124319	KP124174	KP124787	KP125095
	CBS 595.93	KP124320	KP124175	KP124788	KP125096
	CBS 918.96	AF347032	AY278809	KC584435	KC584693
	CBS 102598	KP124329	KP124184	KP124797	KP125105
	CBS 110977	AF347031	AY278808	KC584375	KC584634
	CBS 118811	KP124356	KP124210	KP124824	KP125132
	CBS 118812	KC584193	KC584112	KC584393	KC584652
	CBS 118814	KP124357	KP124211	KP124825	KP125133
	CBS 118818	KP124359	KP124213	KP124827	KP125135
	CBS 121336	KJ862254	KJ862255	KP124833	KP125141
	CBS 121348	KP124367	KP124219	KP124836	KP125144
	CBS 121454	MH863109	AY278812	KP124837	KP125145
	CBS 127672	KP124382	KP124234	KP124852	KP125160
	**IPN 10.0152**	**PP099883**	**PP354051**	**PP354054**	**PP354057**
	**IPN 10.0153**	**PP099884**	**PP354052**	**PP354055**	**PP354058**
*Alternaria arborescens*	CBS 101.13	KP124392	KP124244	KP124862	KP125170
	CBS 105.24	KP124393	KP124245	KP124863	KP125171
	CBS 102605	AF347033	AY278810	KC584377	KC584636
	CBS 750.68	KP124398	KP124250	KP124868	KP125176
	CBS 118389	KP124407	KP124259	KP124877	KP125185
	CBS 119544	KP124408	JQ646321	KP124878	KP125186
	CBS 119545	KP124409	KP124260	KP124879	KP125187
*Alternaria betae-kenyensis*	CBS 118810	KP124419	KP124270	KP124888	KP125197
*Alternaria burnsii*	CBS 108.27	KC584236	KC584162	KC584468	KC584727
	CBS 107.38	KP124420	JQ646305	KP124889	KP125198
	CBS 110.50	KP124421	KP124271	KP124890	KP125199
	CBS 879.95	KP124422	KP124272	KP124891	KP125200
	CBS 118816	KP124423	KP124273	KP124892	KP125201
	CBS 118817	KP124424	KP124274	KP124893	KP125202
*Alternaria eichhorniae*	CBS 489.92	KC146356	KP124276	KP124895	KP125204
	CBS 119778	KP124426	KP124277	KP124896	KP125205
*Alternaria gaisen*	CBS 632.93	KC584197	KC584116	KC584399	KC584658
	CBS 118488	KP124427	KP124278	KP124897	KP125206
	CPC 25268	KP124428	KP124279	KP124898	KP125207
*Alternaria gossypina*	CBS 100.23	KP124429	KP124280	KP124899	KP125208
	CBS 104.32	KP124430	JQ646312	KP124900	KP125209
	CBS 107.36	KP124431	JQ646310	KP124901	KP125210
	CBS 102597	KP124432	KP124281	KP124902	KP125211
	CBS 102601	KP124433	KP124282	KP124903	KP125212
	**IPN 10.0151**	**PP099882**	**PP354050**	**PP354053**	**PP354056**
*Alternaria iridiaustralis*	CBS 118404	KP124434	KP124283	KP124904	KP125213
	CBS 118486	KP124435	KP124284	KP124905	KP125214
	CBS 118487	KP124436	KP124285	KP124906	KP125215
*Alternaria jacinthicola*	CBS 878.95	KP124437	KP124286	KP124907	KP125216
	CBS 133751	KP124438	KP124287	KP124908	KP125217
	CPC 25267	KP124439	KP124288	KP124909	KP125218
*Alternaria longipes*	CBS 113.35	KP124440	KP124289	KP124910	KP125219
	CBS 539.94	KP124441	KP124290	KP124911	KP125220
	CBS 540.94	AY278835	AY278811	KC584409	KC584667
	CBS 917.96	KP124442	KP124291	KP124912	KP125221
	CBS 121332	KP124443	KP124292	KP124913	KP125222
	CBS 121333	KP124444	KP124293	KP124914	KP125223
*Alternaria tomato*	CBS 103.30	KP124445	KP124294	KP124915	KP125224
	CBS 114.35	KP124446	KP124295	KP124916	KP125225
*Alternaria alternantherae*	CBS 124392	KC584179	KC584096	KC584374	KC584633

CBS: Culture collection of the Centraalbureau voor Schimmelcultures, Fungal Biodiversity Centre, Utrecht, The Netherlands; CPC: Personal collection of P.W. Crous, Utrecht, The Netherlands. Accession numbers generated in this study are shown in bold.

## Data Availability

Not applicable.

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
