# Peer review of "Etiology of Foliar Blight of Indian Paintbrush (Castilleja tenuiflora) in Mexico"

_microorganisms, 2024, doi:10.3390/microorganisms12081714_

Round 1

Reviewer 1 Report

Comments and Suggestions for Authors

Dear authors,

I think this paper is fine. I would comment on the technique of pathogen isolation.

Why clorox plant samples to isolate fungi? Were any microbes overlooked this way? What other microbes infected these or related plants before? Was your infection experiment accurately modeling natural infection conditions? Was the natural infection preceded by dry or otherwise unusual weather in the park? Please discuss all the above issues and why you think Alternaris spp. infection was not opportunistic.

Comments on the Quality of English Language

L 39 what does ‘that’ refer to?

L219 ‘produced’ used twice to say the same thing

L220 ‘growth’ used twice to say the same thing

L292 better without comma, with the comma is changes the meaning as if the production of metabolites is not clear

L342 ‘this species’ looks like are fungi. I think you mean the plants. Please re-word

Reviewer 2 Report

Comments and Suggestions for Authors

It is interesting topic, but paper needs to be improved.

Imprecise terminology has been used in lines 41–43: rusts are not Pucciniales, rusts are caused by fungi from the order Pucciniales.

Methods and materials should be described in more detail.

It is strange that only Alternaria spp. were isolated from diseased leaves, usually other fungi are usually found. Authors tentative identified all 24 isolates as Alternaria spp. based on the morphology of conidia. Did all isolate sporulate?

Why were hyphal tips used to obtain axenic isolates? Usually single-spore isolates are used.

Wrong expression “Pathogenicity was performed”.

Several media were used to characterize morphological traits of isolates, but no attempt was made to explain why this was done.

Pictures of IPN 10.0152 and IPN 10.0153 colonies were slightly different, but description (lines 215–225) are common for both isolates. It is unclear.

Discussion is weak, it simply continues introduction about occurrence and importance of Alternaria spp. Results are not discussed in depth, for example – correlations between morphological traits on the different media and genetical diversity. Two isolates of A. alternata are clustered in two clades, but their morphological traits are described as identical.

Authors have not used the latest articles related to phylogeny of Alternaria, including Alternata section, for example, they did not reference the articles by Dettman et al.

Part of conclusion is not based on the results, authors have not studied “disinfestation of the explants to avoid contamination of the vitroplants by some of these pathogens”.

Round 2

Reviewer 1 Report

Comments and Suggestions for Authors

Thanks for heeding my suggestions. Some other points:

L70 This sentence opens a can of worms. What is this material? What is its relation to the diseased plants used to produce data for this paper? Consider removing this sentence.

L331-334 To me, a microbe infects its host, when the host responds by manifesting some markers of infection. For instance, it demonstrates gene-for gene interaction. I would not rule OUT any other type of interaction here, since the data do not allow us to prove that the fungus INFECTS the plant here. On the other hand, the photos of the plants do present some evidence that infection takes place here. Please soften your conclusion

Comments on the Quality of English Language

L321 suceptibility – susceptibility

. 

L355 asseveration – assertion?

Author Response

Reviewer 1

Thanks for heeding my suggestions. Some other points:

Response: Thank you for your time and effort in reviewing our manuscript. Your comments and suggestions have undoubtedly allowed us to considerably improve the article.

L70 This sentence opens a can of worms. What is this material? What is its relation to the diseased plants used to produce data for this paper? Consider removing this sentence.

Response: Thank you for pointing this out. We improve the document according to your suggestion. We delete the sentence.

L331-334 To me, a microbe infects its host, when the host responds by manifesting some markers of infection. For instance, it demonstrates gene-for gene interaction. I would not rule OUT any other type of interaction here, since the data do not allow us to prove that the fungus INFECTS the plant here. On the other hand, the photos of the plants do present some evidence that infection takes place here. Please soften your conclusion

Response: Thank you for pointing this out. We agree that we cannot rule out any other type of interaction, so we have removed the following sentence: “thus ruling out a saprophytic lifestyle”.

L321 suceptibility – susceptibility

Response: Thank you for pointing this out. We have corrected the error

L355 asseveration – assertion?

Response: Thank you for pointing this out. We have corrected the error

Reviewer 2 Report

Comments and Suggestions for Authors

Please, review list of references, text books are not appropriate references.

Conclusions still should be improved. You have not researched  "in vitro propagation is recommended, so attention must be paid to the disinfestation of the explants to avoid contamination of the vitroplants by some of these pathogens". Additionaly, in vitro propogation cannot help to avoid infection with Alternaria spp.

Author Response

Reviewer 2

Conclusions still should be improved. You have not researched  "in vitro propagation is recommended, so attention must be paid to the disinfestation of the explants to avoid contamination of the vitroplants by some of these pathogens". Additionaly, in vitro propogation cannot help to avoid infection with Alternaria spp.

Response: Thank you for your time and effort in reviewing our manuscript. Your comments and suggestions have undoubtedly allowed us to considerably improve the article. Thank you for pointing this out. We agree with you, we did not research in vitro propagation, so we have removed the following sentences:

Line 414: “and limit in vitro propagation”

Line 417:  “so attention must be paid to the disinfestation of the explants to avoid contamination of the vitroplants by some of these pathogens”